# Forecasting the spatial spread of an Ebola epidemic in real time: Comparing predictions of mathematical models and experts

James D Munday[1,2,3]*[†], Alicia Rosello[1,2†], W John Edmunds[1,2†], Sebastian Funk[1,2†]

[1]Centre for Mathematical Modelling of Infectious Diseases, London School of Hygiene and Tropical Medicine, London, United Kingdom; [2]Department of Infectious Disease Epidemiology, London School of Hygiene and Tropical Medicine, London, United Kingdom; [3]Department of Biosystems Science and Engineering, ETH Zürich, Zürich, Switzerland

## eLife Assessment

This article provides **valuable** evidence comparing the performance of mathematical models and opinions from experts engaged in outbreak response in forecasting the spatial spread of an Ebola epidemic. The evidence supporting the conclusions is **convincing**. It will be of interest to disease modellers, infectious disease epidemiologists, policymakers, and those who need to inform policymakers during an outbreak.

## Abstract

**Background:** Ebola virus disease outbreaks can often be controlled, but require rapid response efforts frequently with profound operational complexities. Mathematical models can be used to support response planning, but it is unclear if models improve the prior understanding of experts.
**Methods:** We performed repeated surveys of Ebola response experts during an outbreak. From each expert, we elicited the probability of cases exceeding four thresholds between 2 and 20 cases in a set of small geographical areas in the following calendar month. We compared the predictive performance of these forecasts to those of two mathematical models with different spatial interaction components.
**Results:** An ensemble combining the forecasts of all experts performed similarly to the two models. Experts showed stronger bias than models forecasting two-case threshold exceedance. Experts and models both performed better when predicting exceedance of higher thresholds. The models also tended to be better at risk-ranking areas than experts.
**Conclusions:** Our results support the use of models in outbreak contexts, offering a convenient and scalable route to a quantified situational awareness, which can provide confidence in or to call into question existing advice of experts. There could be value in combining expert opinion and modelled forecasts to support the response to future outbreaks.

**Funding:** This study was partly funded by the Department of Health and Social Care using UK Aid funding 47 and is managed by the National Institute for Health and Care Research (VEEPED: PR-OD-1017- 48 20002; AR and WJE). This study was partly funded by the Wellcome Trust (210758/Z/18/Z : JDM 49 and SF). The views expressed in this publication are those of the authors and not necessarily 50 those of the funders.

*For correspondence:
james.munday@swisstph.ch

[†]These authors contributed equally to this work

Competing interest: The authors declare that no competing interests exist.

**Figure 1.** The extent of the 2018–2020 Ebola outbreak in the northeastern Democratic Republic of the Congo (DRC) and areas included in our study. (**A**) Daily incidence in the northeastern DRC between August 2018 and March 2020. Grey points show days prior to the study period, coloured points show days within the study period (November 2019–March 2020), and hue indicates month. (**B**) shows the total number of cases of Ebola recorded in each health zone. (**C**) Number of cases in each month and health zone during the period covered by this study; health zones outlined in red show all health zones affected by the entire epidemic.

## Introduction

Following the initial emergence in 1976 in Zaire (now the Democratic Republic of the Congo [DRC]) (*Report of an International Commission, 1978*), epidemics of Ebola virus disease (EVD) have occurred, on average, every 12–24 months (*Rosello et al., 2015*). EVD is a viral haemorrhagic fever first caused by the Ebola Zaire virus (EZV), with a case fatality rate of 25–90% (*In press, 2023b*). A major outbreak in the northeastern provinces of DRC between 2018 and 2020 resulted in over 3300 reported cases and over 2100 deaths (*WHO Regional Office for Africa, 2020*; *Figure 1*).

Transmission of EZV occurs mainly through direct contact during the symptomatic phase of infection; therefore, isolation of infected individuals with strict infection control, contact tracing, and safe burials has been key to controlling past EVD outbreaks (*Kucharski et al., 2015*), although setting specific challenges can hamper containment efforts (*Adongo et al., 2016*). More recently, vaccination has also become a tool for outbreak control, with two vaccines now licensed for use (*In press, 2023a*; *Woolsey and Geisbert, 2021*).

EVD outbreaks typically occur in resource-poor settings where limited communication and poor accessibility make logistics of surveillance and vaccination campaigns challenging. Understanding the spatial risk of future spread is therefore useful to allow response teams to focus efforts on high-risk areas. Mathematical and statistical models have been used extensively to forecast the spread of

infectious diseases, including EVD (*Chowell et al., 2017*). Such models rely on a combination of statistical inference based on epidemiological data and information about the mechanisms underlying the dynamics of infection. However, the dynamics of EVD are frequently governed by changing contextual factors which are challenging to forecast quantitatively. For example, violent conflicts or flooding can seriously hinder, interrupt, or even reverse the impact of containment efforts (*Adongo et al., 2016*; *Wannier et al., 2019*). Moreover, changes in healthcare capacity and health-seeking behaviour of patients can strengthen or weaken efforts to reduce transmission (*Funk et al., 2017*). The timing and impact of these factors are notoriously difficult to predict using mathematical models.

Models are used by epidemic response experts to support decision-making in the field. In addition to models, experts also make judgements as to the future spread of the virus based on their interpretation of the current status of the outbreak combined with their knowledge of other less tangible factors such as the geography, climate (e.g. seasonal variation in accessibility of particular areas), and soft intelligence about the escalation of conflict in areas which may, as a result, be harder to access by response teams. There are clear costs and benefits to human-made and model-based forecasts. Whereas models are objectively based on observations of the past outbreak dynamics and current case data, experts have additional knowledge of the complex factors surrounding the outbreak response. It is therefore difficult to assess the impact mathematical models have on decision-making, how much modelled forecasts differ from those made by experts in the field, and whether either modelled or human forecasts are systematically more accurate or useful. Moreover, the knowledge of experts in the field of EVD epidemiology, with a good understanding of the geographical area of study, may provide an invaluable resource that is currently underused in forecasting.

Previous studies have aimed to establish the relative performance of humans and models in predicting infectious disease spread in human populations, particularly in the context of acute respiratory infections such as influenza and SARS-CoV-2. Three studies have evaluated the predictions of humans against models explicitly. The first of these evaluated short-term forecasts and season-wide predictions of reports of influenza-like illness in the United States (*Farrow et al., 2017*) and two studies (*Bosse et al., 2022*; *Bosse et al., 2023*) compared short-term forecasts of cases of and deaths from COVID-19, firstly in Germany and Poland and secondly in the United Kingdom. All three studies found that humans tended to perform better than the mathematical and statistical models selected for comparison when predicting cases. However, the COVID-focused studies found that the human ensembles performed worse than the ensemble prediction of the models when predicting deaths. These results were maintained when only self-declared 'experts' were included in the forecasts. A number of other studies recorded expert predictions without comparison to mathematical models. A study conducted early in the COVID-19 pandemic (*Recchia et al., 2021*) evaluated the relative ability of laypeople and experts to predict the course of the UK epidemic over the first calendar year. The study found that both experts and laypeople typically underpredicted the impact overall; however, experts' forecasts were more accurate and better calibrated than laypeople. A study of expert predictions in the United States (*McAndrew and Reich, 2022*) evaluated their weekly forecasts of case incidence and total deaths in the first year against a pooled ensemble of all predictions. The study found that the ensemble outperformed every expert individually over the period of the study. A similar study surveyed experts regarding the total number of cases and deaths from MPox in the United States during 2022 (*McAndrew et al., 2022*); however, these predictions are yet to be evaluated. Overall, these studies provide evidence that human predictions can play a valuable role in epidemiological prediction, providing a comparator and complementary method to mathematical and statistical modelling.

In this article, we extend the use of expert forecasters to predict spatial risk of transmission in the context of a local outbreak. We made monthly forecasts of the geographic spread of EVD from November 2019 to March 2020 during the declining phase of the 2018–2020 outbreak in the DRC using both expert predictions collected through regular interviews and with two spatially explicit computational transmission models with different spatial interaction assumptions: a gravity model and an adjacency model (where transmission can only occur between contiguous regions; see the 'Methods' section for details). Alongside supporting situational awareness, these forecasts were motivated by an aim to inform site selection for a planned vaccine trial. The objective was to identify areas that had seen no cases yet and thus were not already being supported by vaccination and other interventions, but were at high risk of still becoming affected by the EVD outbreak, thus allowing

estimation of efficacy (*Watson-Jones et al., 2022*). Here we evaluate the performance of the forecasts and select ensembles of the methods in predicting continued transmission and flare-ups of EVD in health zones (HZs) close to the affected area. We further study variation in forecast quality against a selection of factors related to local demography, case history, and forecast implementation.

## Methods

### Expert elicitation

Experts in EVD epidemiology with knowledge of the local geography speaking English or French were identified originally by convenience sampling. The pool of experts was then expanded through recommendation from the identified experts (snowball sampling). This approach was best suited to capture the expertise of individuals who were most often temporarily based in the field.

A pilot study was carried out in November 2019. Subsequently, monthly interviews were held over WhatsApp in December 2019, January 2020, February 2020, and March 2020. All interviews were scripted. The main biases of this type of study (availability bias, representativeness bias, overconfidence, motivational bias, anchoring on past estimates) were briefly discussed during the first interview.

Experts were also provided with an interactive map of the outbreak area and surrounding HZs showing the number of total cases during the outbreak and during the two preceding weeks for reference (*Figure 3—figure supplement 1*). HZs were numbered to facilitate communication with the experts.

Experts were asked to estimate the number of reported probable and confirmed cases they would expect per HZ during the following month using the online MATCH Uncertainty Elicitation Tool (*Morris et al., 2014*; *Figure 3—figure supplement 1*). Through this platform, the experts and the researcher (AR) interacted in real time. The 'roulette' (chips and bins) method was used. Experts were instructed to place a total of 20 chips over the available bins (0–1 cases, 2–4 cases, …, 48–50 cases). Therefore, each chip represented for the expert a 5% probability that the number of cases was in the bin where the chip was placed. This process aimed to capture the uncertainty surrounding the expert's estimates.

The experts were asked to estimate the number of reported cases they would expect in the HZ where there had been one or more cases in the two preceding weeks, as well as Goma (*Figure 2*).

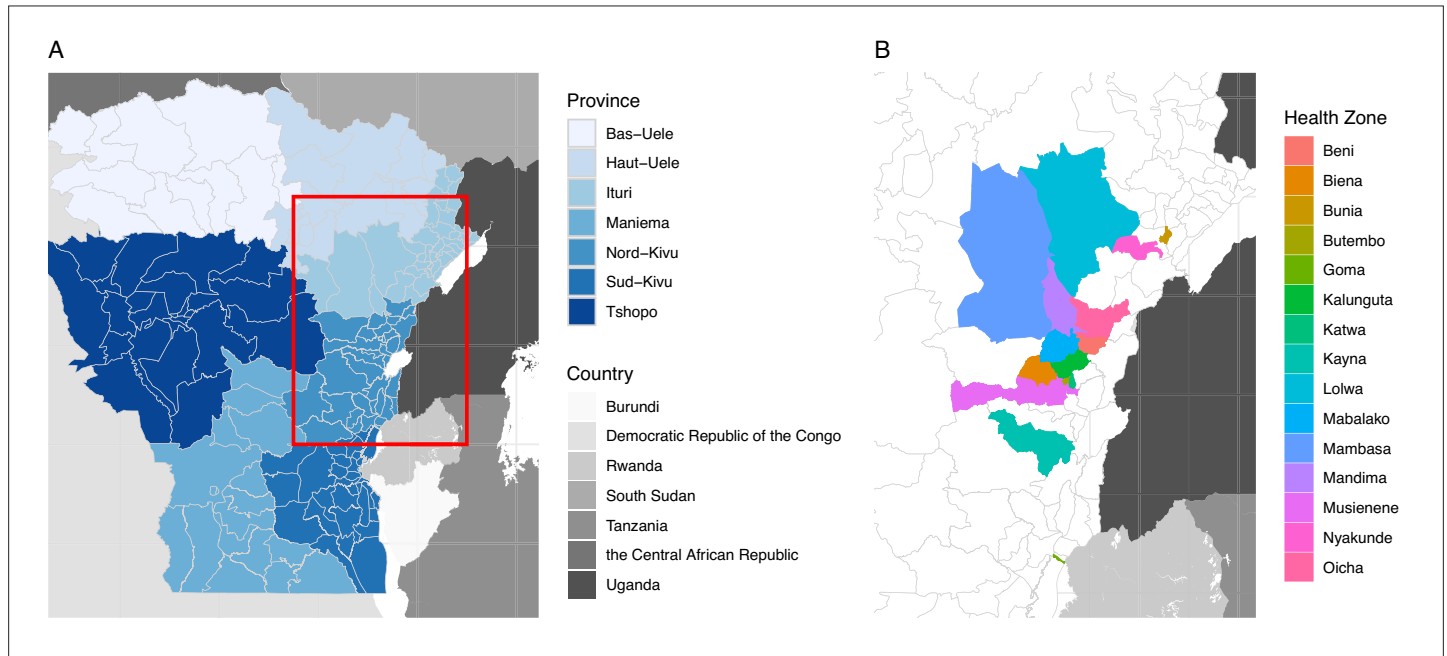

**Figure 2.** Health zones included in the model and expert elicitation survey. (**A**) shows the provinces around the affected area, and included in the transmission model, and the red box shows the area detailed in panel (**B**). (**B**) shows the health zones included at least once in the expert elicitation survey we conducted.

The experts were then asked to identify any additional HZ where they would predict one or more cases during the following month with >5% probability, and to also estimate the number of reported cases they would expect in these HZ. In the pilot study, carried out in October 2019, experts were asked to forecast the number of cases they expected during November 2019 in 10 HZs: Beni, Goma, Kalunguta, Katwa, Lolwa, Mabalako, Mambasa, Mandima, Nyankunde, and Oicha.

## Modelling framework

In parallel with the expert elicitation programme, we developed a modelling framework to forecast spatial risk of infection. In the framework, the incidence of cases is forecast in each HZ based on historical case reports. The model was formed of two components, the autoregressive component and the spatial component.

$$\lambda = \sum_{s=t-(D+L)}^{t-D} \left( \gamma N_{i,s} + \alpha \sum_j w_{ij} N_{j,s} \right) \tag{1}$$

The auto-regressive component modelled the rate of infections in a particular HZ $i$, on day $t$, to be proportional to the number of cases in the same HZ ($i$) between dates $t$-$(D+L)$ and $t$-$D$, where $L$ is the estimated latent period and $D$ is the estimated infectious period. The spatial component accounts for transmission between HZs, where rate of infection was proportional to the cases in each other HZ (i.e. $\forall j\, j{\neq}i$) and moderated by a pairwise specific factor defined by a spatial kernel $w_{ij} w_{ij} w_{ij}$. We used two spatial kernels, both of which use proximity of HZs to each other and their respective population size, $w_{ij} P_i P_i P_i$ and $P_i P_j P_j P_j$. Firstly, the gravity model which treats interaction in an analogous way to Newtonian gravity with population size in place of mass, such that interaction reduces distance, $P_j d$, raised to a power, $k$.

$$w_{ij} = \frac{P_i P_j}{d_{ij}^k} \tag{2}$$

Secondly, we applied a model with adjacency-based interaction. In this model, only adjacent HZs can interact. The strength of interaction between HZs is proportional to the product of their population sizes.

$$w_{ij} = \delta_{ij} P_i P_j \tag{3}$$

$$\delta_{ij} = \begin{cases} 1, \text{if adjacent} \\ 0, \text{otherwise} \end{cases} \tag{4}$$

Cases were modelled as Poisson distributed such that

$$N_{i,t} = \text{Poisson}\left(\lambda_{i,t}\right) \tag{5}$$

To forecast cases, we fitted the spatiotemporal model to historical data from the 60 days prior to the date the forecast was made, accounting for cases in HZs in seven regions (169 HZs) centred on the location of the epidemic; Nord-Kivu, Ituri, Tshopo, Maniema, Sud-Kivu, Haut-Uele, and Bas-Uele. We fit the model using the No U-Turn Sampling (NUTS) method for Hamiltonian Monte Carlo with Stan (*Stan Team, 2012*), a probabilistic programming framework. We estimated $\alpha$ and $\gamma$, which vary the contribution of within-health-zone and between-health-zone transmission. We also estimated $k$, which determines how rapidly transmission rate decays with distance in the spatial component of the model. We sampled parameters from the resultant joint posterior distribution to simulate daily incidence in all HZs in the seven regions, up to and including the last day of the following month. We performed 1000 iterations for each forecast date. We then extracted the full distribution of the number of cases incident within the calendar month of interest. Forecasts were made using data up to the last day of the month prior to the forecast period.

## Ensemble forecasts

Ensemble forecasts were calculated as an average of the probabilities attributed by the members of the ensemble. For the expert ensemble, the arithmetic mean was calculated across all experts with equal weighting. Similarly, the model ensemble used the unweighted mean of the model forecasts. For the mixed (model and expert) ensemble, the mean was weighted such that the combined weight of the experts' forecasts and the combined weight of the models' forecasts were equal.

## Quantification of risk and forecast evaluation

To compare the model and the expert forecasts and score them according to the eventual true number of cases, we calculated the probability attributed to cases over four thresholds, ≥2, ≥6, ≥10, and ≥20 cases.

We evaluated the forecasts using the Brier score, a proper scoring rule which quantifies how accurate a forecast or a group of forecasts is when compared to true data after the event. The Brier score, BS, is defined as the square of the difference between the probability of observing an event and the observation $o_i$ status, which takes a value 1 or 0 for cases observed and none observed, respectively. We calculated this for multiple ($N$) forecasts by taking the mean of the individual forecast scores.

$$BS = \frac{1}{N} \sum_{i=1}^{N} \left( p_i - o_i \right)^2 \tag{6}$$

We also quantified the general bias and calibration of the forecasts by considering the hazard rate predicted by each forecast, which we calculated as the sum of probabilities attributed to exceeding each threshold. This gives the number of HZs the forecast 'expected' to cross the threshold in each month. To quantify the bias of each set of predictions, we took the difference between the hazard rate and the actual number of HZs that exceeded each threshold in each month. We refer to this as the *hazard gap* (HG).

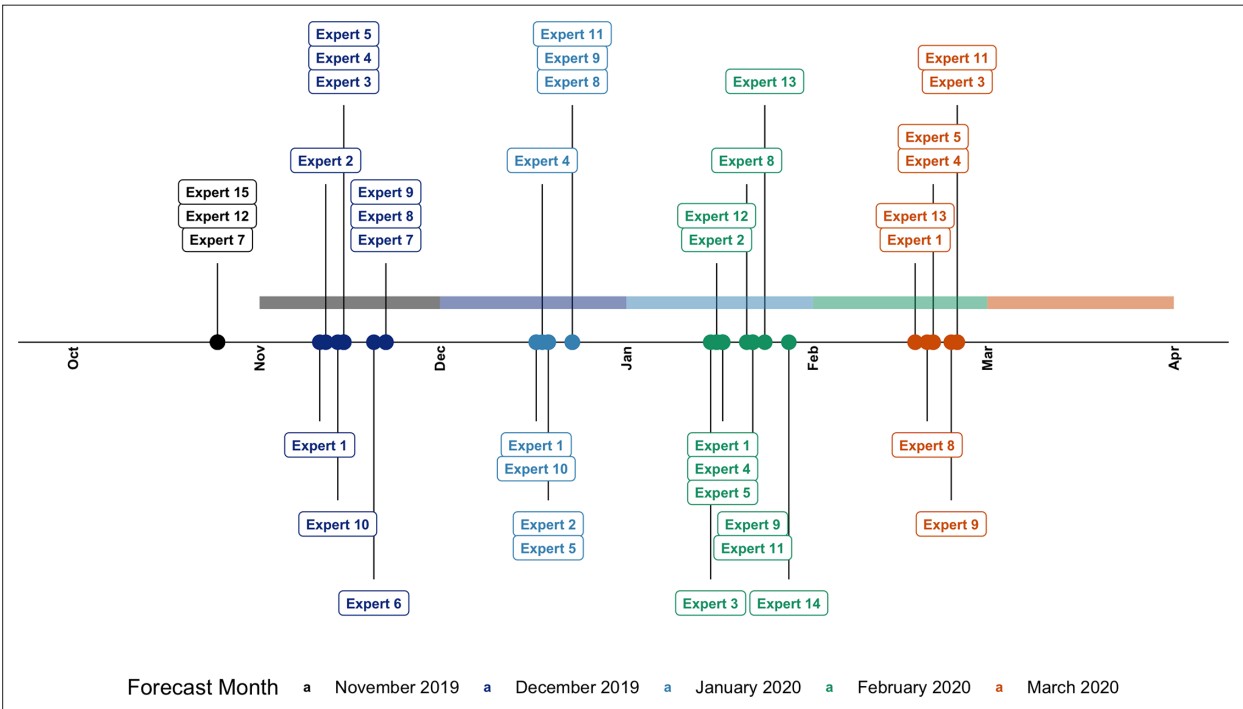

**Figure 3.** Timeline of the expert elicitation. Each point shows the date of the interview of the expert labelled to obtain forecasts for the following month. Colour indicates the month for which the forecast was made, and the forecast windows are highlighted with a shaded band of the same colour.

The online version of this article includes the following figure supplement(s) for figure 3:

**Figure supplement 1.** Tools used for expert elicitation.

$$HG = \sum_{\forall HZ} p\left(c > c_{thresh}\right) - \sum_{\forall HZ} o\left(c > c_{thresh}\right) \tag{7}$$

where $p(c>c_{thresh})$ is the probability of threshold exceedance and $o(c>c_{thresh})$ is the observation status of the threshold exceedance 1 if true 0 if false.

## Results

### Expert panel and HZs included in survey

Over the study period, we conducted a total of 40 interviews with 15 experts, 3 of which took place during the pilot phase (November 2019). *Figure 3* shows the timeline of the expert elicitations.

Eight experts worked at the World Health Organization, four at the London School of Hygiene & Tropical Medicine, two for Médecins Sans Frontières, and one at the DRC Ministry of Health. Most experts (10/15) had more than 5 years of experience working in infectious disease epidemiology. About half of the interviews (21 of 40) were conducted with experts who were in the outbreak area (defined HZs affected by EVD or Goma, the site of the international response base) or had been there within 2 weeks of the interview. Four experts had never been in the outbreak area.

Between December 2024 and March 2025, 8–10 experts provided monthly forecasts for 4–11 HZ, estimating the probability that reported cases would exceed various thresholds (2, 6, and 20 cases) (*Figure 4—figure supplements 1–4*). Expert predictions showed variable accuracy across months and zones.

Experts correctly identified Mabalako as the highest-risk HZ in December. They attributed an average 82% probability of exceeding 2 cases; Mabalako reported 38 cases that month, exceeding all thresholds, although the probability assigned to exceeding the higher thresholds was similar to that of Beni (3 cases). Other zones with moderate cases—Kalanguta (5) and Mambasa (4)—were assigned lower probabilities and generally did not exceed higher case thresholds. Interestingly, Mandima, which had no reported cases, was assigned a relatively high probability (72%) of exceeding two cases, indicating some overestimation in this zone.

In January, Beni and Mabalako again accounted for most cases (22 and 11, respectively) and were recognized as high risk by the experts. However, experts underestimated the scale of the outbreak in Beni, assigning 0% probability to exceeding 20 cases there, despite this threshold being surpassed. Mandima continued to be forecasted as high risk, though no cases were reported. Predictions for other zones such as Oicha and Biena suggested moderate risk, but no cases were confirmed.

Of the 11 nominated HZs, only Beni reported confirmed cases in February (9), exceeding the six-case threshold. Experts collectively assigned a 70% probability to this event. Similar probabilities were assigned to Mabalako (60%), where no cases were reported, following cases in the prior 2 months. Several other zones were predicted to exceed two cases, yet reported none.

No cases were reported in any HZ during March. Experts broadly anticipated this, with only one expert assigning over 50% probability of exceeding two cases in any HZ. Beni had the highest average assigned risk at 33% (*Figures 4 and 5*).

### Performance evaluation

We evaluated the forecasts using the Brier score. The overall scores of individual experts varied between 0 and 0.6 across the four thresholds. Collectively, the experts scored best at the highest threshold (20 cases) and worst for forecasts of the lowest threshold (2 cases). The models also performed better at higher thresholds than low thresholds, but the difference was less pronounced. Overall, the gravity model ranked best among all forecasts at the two-case threshold. It also ranked best for this threshold in the month of February and consistently in the top half of forecasts in December and January; however, it performed comparatively poorly in March, ranking higher than only one of the experts. The adjacency model also performed better than the experts overall for the two-case threshold. Related to this, including the models improved the ensemble forecast. Although the gravity model performed better than the adjacency model for higher thresholds, together the models performed similarly to the expert ensemble forecast overall. In January and February, the gravity model performed well compared to the adjacency model and the expert ensemble; however in March, both models performed particularly badly compared to the experts

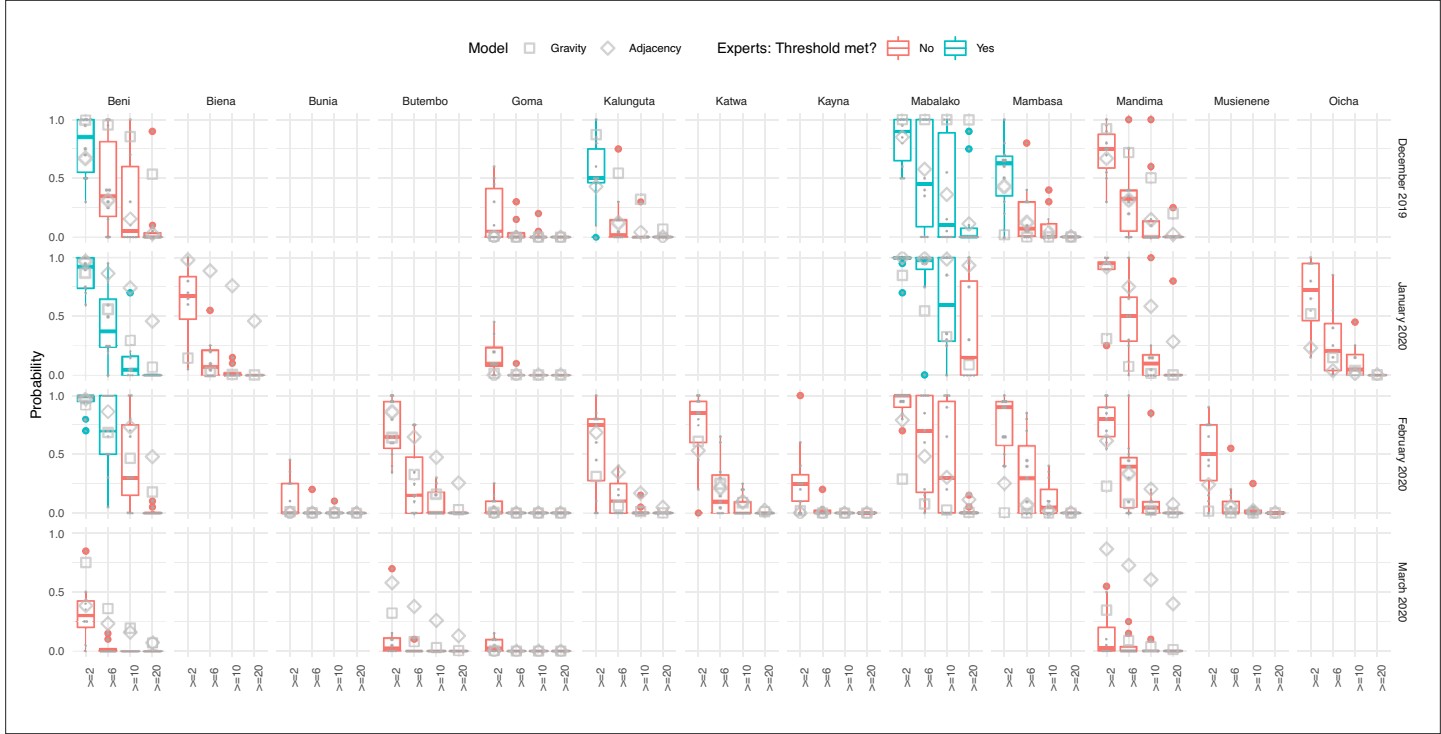

**Figure 4.** Expert elicitation results and accuracy of predictions. Only the health zones (HZs) that were rated by all experts are included here. Results are shown as probabilities (vertical axes) that a given HZ (horizontal panels) exceeds a given threshold (horizontal axes) according to the experts (box plots) or models (square/diamond for gravity and adjacency models, respectively) across different months (vertical panels). HZ/month combinations where the given thresholds were exceeded are marked in cyan, and ones where they were not are marked in red.

The online version of this article includes the following figure supplement(s) for figure 4:

**Figure supplement 1.** Expert forecasts made using MATCH for the distribution of cases expected in the month of December.

**Figure supplement 2.** Expert forecasts made using MATCH for the distribution of cases expected in the month of January.

**Figure supplement 3.** Expert forecasts made using MATCH for the distribution of cases expected in the month of February.

**Figure supplement 4.** Expert forecasts made using MATCH for the distribution of cases expected in the month of March.

for all thresholds. None of the experts performed consistently well relative to the others; experts 3 and 10 performed best for the two-case threshold, whereas experts 13 and 14 did best for higher thresholds.

To evaluate how the different forecasts may impact decision-making, we ranked the HZs for each month, based on the probability of exceeding each threshold of cases forecast by each ensemble and by the model alone (*Figure 5—figure supplement 1*). In general, the model and the ensembles all ranked HZs that did reach the threshold highly. In some cases, the model performed better, ranking HZs that did meet the threshold higher than the experts, specifically ranking Beni higher than Mandima in higher thresholds (≥ 6 and 10 cases) for the forecast of January, where Beni ultimately had cases and Mandima did not, in that month. Considering the models separately, the gravity model performed better than the adjacency model in general, with the adjacency model occasionally performing worse than the experts when ranking the HZs. This was clearest in the forecasts of November and January.

## Bias and calibration in forecasts

We evaluated the bias in each forecast type by considering the hazard gap between forecasts and actual cases. We found that experts systematically forecasted higher risk of the lowest threshold (≥2 cases) than was warranted, but tended to forecast lower risk of exceeding the highest threshold (≥20 cases) than was borne out across all HZs (*Figure 6*). When calculated across all months, this bias was present in 12 of the 15 experts. The models did not show clear, consistent bias in either direction.

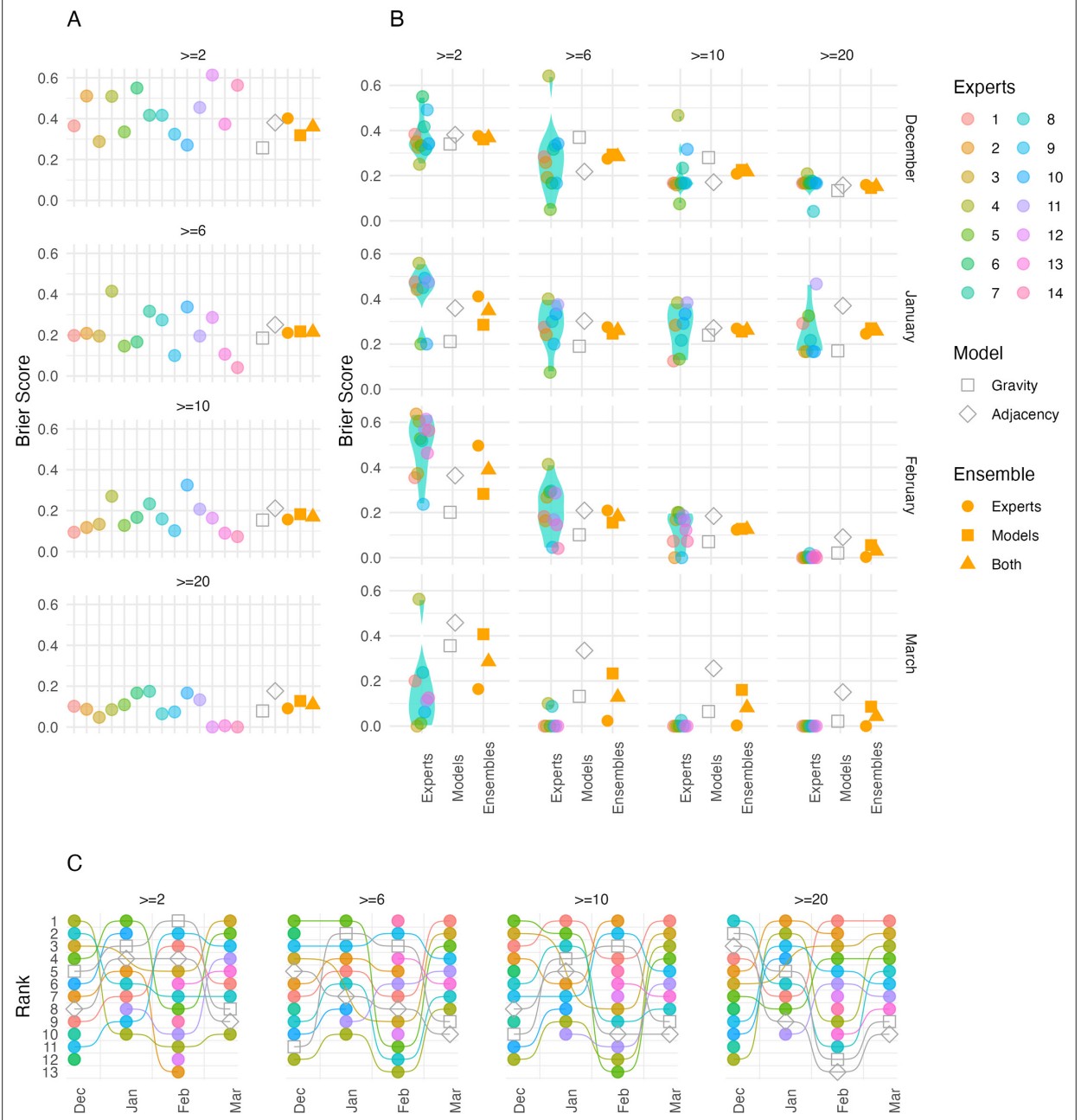

**Figure 5.** Evaluation of forecasts made by the experts, models, and ensembles. (**A**) shows the overall Brier score for each expert, model, and ensemble (calculated over all forecasts included in the study). In (**B**), each panel shows the Brier score across all health zones for each month (vertical) at each case threshold (horizontal). Coloured points show each expert score, and the violin plot shows their distribution. The grey hollow points show the model scores, the yellow points show the ensemble scores (circles show experts alone, squares show models alone, and triangles show experts and models with 50% weight given to each). (**C**) shows the ranking of each expert and model in terms of forecast performance.

The online version of this article includes the following figure supplement(s) for figure 5:

**Figure supplement 1.** Risk rank of health zones.

## Forecasting flare-ups

In addition to the HZs presented to all experts, each expert was able to nominate HZs, which they deemed at risk. Experts nominated seven further HZs to forecast in December, four in January, four in February, and one in March (*Table 1*).

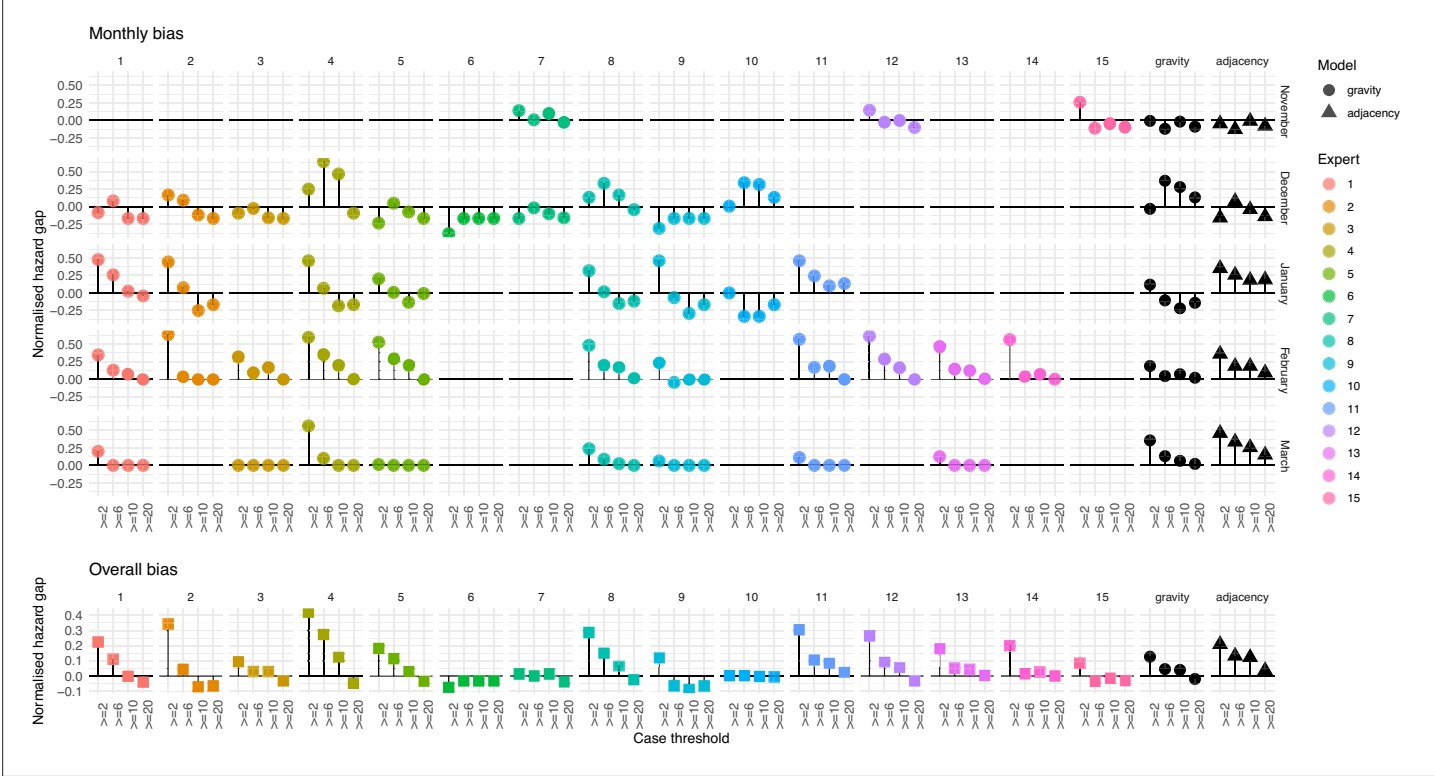

**Figure 6.** Bias and calibration of forecasts. Panels show the hazards gap difference between the hazard rate (expected number of exceedances across all health zones) for each threshold attributed by the forecast and the actual number of health zones that exceeded the associated threshold. Each panel shows one forecast (expert or model) in each month. The bottom row shows the same for each forecast calculated over the entire study period.

The only two HZs not included in the default list but exceeding two cases were Butembo and Katwa. These were nominated by 2 of the 10 experts (4 and 10), both attributing 50% chance of exceeding two cases. In contrast, Oicha and Komanda had the most nominations with 7 and 5 each, and 6 and 4 of the 10 experts interviewed allotted greater than 5% chance of two or more cases in December, with attributed probabilities ranging between 5% and 95% chance of crossing the two-case threshold.

In January, six of the eight experts interviewed nominated Butembo (1, 2, 4, 5, 9, and 11 with probabilities of 5–85% of crossing the two-case threshold) while three of them also nominated Katwa (2 and 9 giving 85% and 5 giving 5%). Kalunguta and Manguredjipa were also nominated by one expert each, 5 with 10% and 11 with 50%, respectively. None of the nominated HZs crossed the threshold in January.

In February, 6 of the 10 experts interviewed (2, 4, 5, 11, 12, and 14) nominated Oicha with probabilities between 10% and 95% of crossing the two-case threshold. Four (2, 4, 5, and 11) nominated Biena with probabilities between 10% and 95% of exceedance. Experts 4 and 8 also nominated Vuhovi, attributing 55% and 20% probability of threshold exceedance, respectively. Expert 3 nominated Lolwa alone but gave no probability of exceeding one case. No HZs not included in the interview as default crossed the two-case threshold in February.

In March, three (4, 8, and 11) of the eight experts interviewed gave probabilities of 35%, 50%, and 15% of exceeding the two-case threshold, respectively. No HZs not included in the interview as default crossed the two-case threshold in March.

To compare the model with the experts, we included all HZs modelled and attributed all HZs not nominated by experts an exceedance probability of 0%. To allow comparison, we also set all HZs given a probability of lower than 5% to 0% for both the gravity and adjacency models. When considering the Brier score (*Figure 7*), we found that the gravity model performed comparably to some experts when forecasting for December and February. The adjacency model performed worse than all

**Table 1.** Experts and health zones included in each round of the survey.

The left part of the table details the experts interviewed (highlighted in green) the health zones included in the main survey in each month. In addition, the right part of the table details the health zones nominated by experts and the number of experts that nominated each one.

| Month forecasted | Experts interviewed (highlighted) | | | Health zones in interview (HZs) | No. of experts | HZ nominated |
|---|---|---|---|---|---|---|
| December | 1 | 2 | 3 | Beni | 7 | Oicha |
| | 4 | 5 | 6 | Goma | 5 | Komanda |
| | 7 | 8 | 9 | Kalunguta | 2 | Butembo |
| | 10 | 11 | 12 | Mabalako | 2 | Katwa |
| | 13 | 14 | | Mambasa | 1 | Lolwa |
| | | | | Mandima | 1 | Makiso-Kisangani |
| | | | | | 1 | Nyankunde |
| January | 1 | 2 | 3 | Beni | 6 | Butembo |
| | 4 | 5 | 6 | Biena | 3 | Katwa |
| | 7 | 8 | 9 | Goma | 1 | Kalunguta |
| | 10 | 11 | 12 | Mabalako | 1 | Mangurerdjipa |
| | 13 | 14 | | Mandima | | |
| | | | | Oicha | | |
| February | 1 | 2 | 3 | Beni | 6 | Oicha |
| | 4 | 5 | 6 | Bunia | 4 | Biena |
| | 7 | 8 | 9 | Butembo | 2 | Vuhovi |
| | 10 | 11 | 12 | Goma | 1 | Lolwa |
| | 13 | 14 | | Kalunguta | | |
| | | | | Katwa | | |
| | | | | Kayna | | |
| | | | | Mabalako | | |
| | | | | Mambasa | | |
| | | | | Mandima | | |
| | | | | Musienene | | |
| March | 1 | 2 | 3 | Beni | 3 | Mabalako |
| | 4 | 5 | 6 | Butembo | | |
| | 7 | 8 | 9 | Goma | | |
| | 10 | 11 | 12 | Mandima | | |
| | 13 | 14 | | | | |

the experts in every month except February. In every month, the ensemble of experts did better than the models and including the models in the ensembles reduced performance.

## Discussion

We compared forecasts of the geographic spread of Ebola made by experts with those made using a modelling framework. Since the outbreak dynamics of Ebola are highly sensitive to the changeable context in which they take place, mathematical models and expert opinions are expected to have different strengths and weaknesses, with models benefiting from objective inference from previous observations and experts able to utilize detailed knowledge about the outbreak and the changing

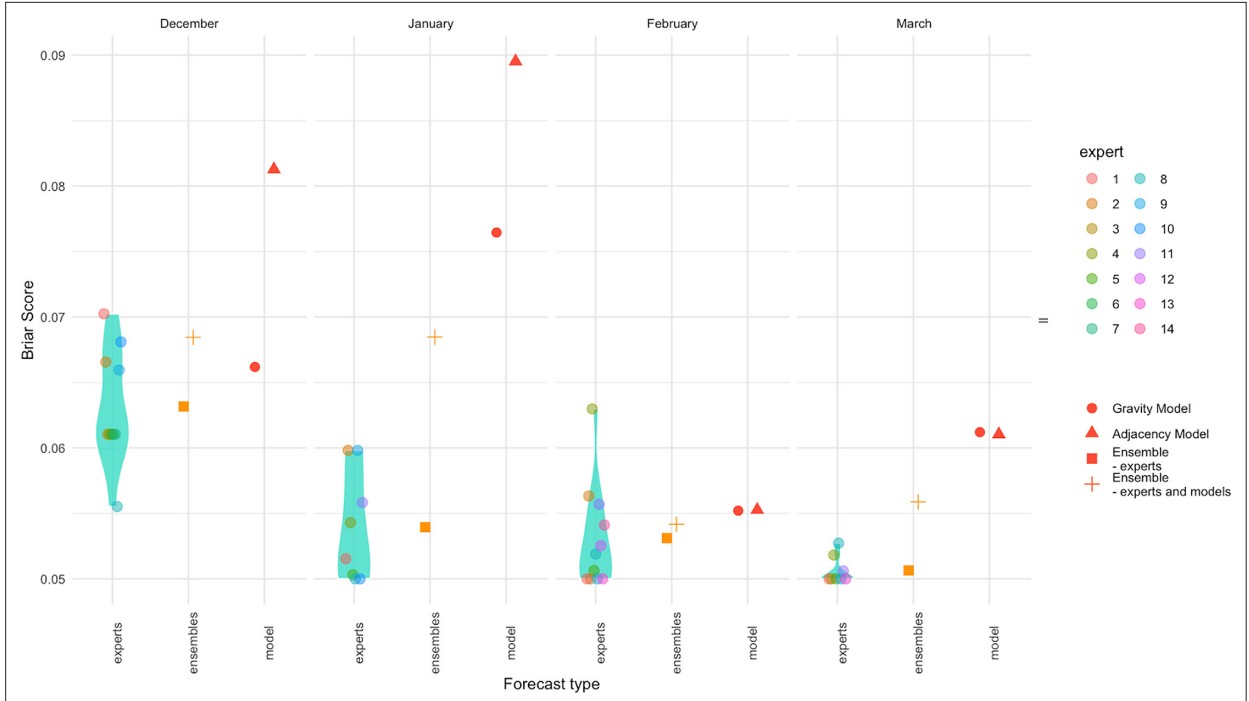

**Figure 7.** Evaluation of forecasts made in health zones not included in the main survey. Each panel (right to left) shows the Brier score across all health zones for each month. Coloured points show each expert score, and the density plot shows their overall distribution. The red points show the model scores, and the yellow points show the ensemble scores (squares show experts alone, and crosses show experts and models with 50% weight given to each).

surrounding context to make informed projections of risk. By interviewing experts and asking them to forecast risk in a structured way, we were able to compare the performance of their forecasts against those made with well-established modelling approaches in a quantifiable and robust way.

Overall, the forecasts made by the group of experts as a whole performed similarly to those of the model, with a few consistent exceptions. The model performed better than the experts when considering the lowest threshold in 4 of the 5 months covered by the survey, but performance was more comparable for the higher thresholds. The model also performed marginally better when ranking HZs by risk of ongoing transmission, indicating that use of a model may improve prioritization of HZs when attributing resources.

We found that both methods performed better when considering higher case thresholds. This is likely to be due to a combination of bias in both forecast types towards under-prediction of cases and the fact that there were few instances where the higher thresholds were reached.

Although individual experts frequently outperformed ensembles in individual instances, no individual expert outperformed the ensembles overall. This supports the practice of considering predictions from a range of experts over a smaller number of more specialist or experienced experts. The models tended to perform similarly to the ensemble representing more consistent performance across all forecasts.

Experts tended to be more biased than models, especially at low case thresholds with a tendency to over-predict cases to a greater degree than models. This bias reduced rapidly as the case threshold increased. This may be interpreted as over-cautiousness from the experts regarding the potential for geographic spread of the virus but confidence that transmission could be contained quickly. This trend reflects a pattern among previous introductions into new HZs earlier in the pandemic, where a small number of cases were reported, but the local outbreak was quickly stopped (*Figure 1*).

To our knowledge, our study is the first to record experts' assessment of geographical risk at a local level during an epidemic and the first comparison of outbreak response experts' predictions to those of models in real time. Although direct comparison is not possible, our results lead to conclusions that are broadly similar to those from previous studies (*Farrow et al., 2017*; *Bosse et al., 2022*;

*Bosse et al., 2023*); however, each of these studies found that ensemble expert forecasts performed better than the comparison models, whereas our study found no clear performance difference. This may suggest that experts are better at predicting simple time series than geographic distribution of cases. However, we cannot view these findings independently from the different survey designs or study contexts.

There are a number of important limitations to consider when interpreting our results. The context within which we conducted the study has important implications for interpretation. Due to the timing and logistics of setting up the questionnaire, the study only began in the closing phase of the epidemic, whereas the relative performance of experts and models may differ during different phases of the epidemic. For example, in the early phase where dynamics are driven more by infectious transmission than established response practices, or during the peak where changes in intervention strategy may be more influential. The stage of the epidemic also meant that there was a substantial trend towards 'negative' results (i.e. no threshold exceedance), which is likely to favour some forecasting methods over others.

Additionally, experts were not all interviewed on the same day, and interviews occurred several days before the beginning of the month they were forecasting. In some cases, experts were interviewed up to 2 weeks prior to the beginning of the month. This means that the information available differed both between experts and with the model, which was run considering data up to the first day of each month. This reduces our ability to compare models to experts directly; however, it could also be argued that this is a 'built-in' factor which represents the inherent challenge of eliciting predictions from experts. In addition, the interview process for experts was quite taxing and required a phone call, which can cause scheduling challenges during an epidemic response. It may be that other methods with less arduous and more flexible data entry would improve responses.

Our analysis represents the comparison of expert forecasts to only two specific forecasting models. There is a great range of models that could have been applied in this context which may have differed in performance to those we used. We chose these models for convenience since we were applying them to the outbreak at the time of the interviews. It is also possible that some of the experts involved in the study had ingested results from our model, which were available through our online dashboard, or other models being used at the time.

Since our findings, like those of similar studies, suggest that models and experts perform comparably in this context, there is an argument that models have no value in informing expert decision-making. It can be argued, however, that models remain useful in outbreak response. Firstly, while the models performed similarly to the ensemble forecasts of the experts, there was no individual expert that performed consistently better than the models. Secondly, models are much more easily scaled and generalized, making them simple to deploy in new contexts and to adapt as epidemics grow. Expert interviews are time-consuming and often inconvenient, especially in the context of outbreak response activities, which are characteristically fast paced. Models therefore offer a more convenient route to a quantified insight, which from our results performs comparably to the way groups of experts may think. Finally, there are ways to combine both methods. For example, in the event that expert forecasts can be garnered, joint ensembles can capture information from both the expert and modelled forecasts. Further, we suggest that models can offer a role in aiding decision-making by providing confidence in or calling into question expert advice that is being considered.

## Conclusions

Our analysis evaluated performance of experts and models when forecasting the spatial spread of Ebola, representing the first such study incorporating local geographic distribution and the first to focus on an epidemic in a resource-poor setting. We found that forecasts made by experts and models performed comparably overall, but experts tended to be slightly more biased towards predicting that a small number of cases would persist. The results support the use of models in outbreak response and provide insight into how models and expert opinion could be combined when tackling future epidemics.

## Acknowledgements

We thank Xavier de Radiguès, Neale Batra, Nabil Tabbal, Mathias Mossoko, Chris Jarvis, Thibaut Jombart, Denis Ardiet, Michel Van Herp, Silimane Ngoma, Olivier le Polain, Esther van Kleef, Noé

Guinko, and Amy Gimma as well as two other experts who preferred to remain anonymous, for their participation as experts in this study. We would also like to thank David Smith, Thibaut Jombart, Chris Jarvis, Flavio Finger, and Anton Camacho for their helpful advice in conducting this survey. This study was partly funded by the Department of Health and Social Care using UK Aid funding and is managed by the National Institute for Health and Care Research (VEEPED: PR-OD-1017-20002; AR and WJE). This study was partly funded by the Wellcome Trust (210758/Z/18/Z: JDM and SF). The views expressed in this publication are those of the authors and not necessarily those of the funders.

## Additional information

### Funding

| Funder | Grant reference number | Author |
|---|---|---|
| Wellcome Trust | 10.35802/210758 | James D Munday<br>Sebastian Funk |
| National Institute for Health and Care Research | VEEPED: PR-OD-1017-20002 | Alicia Rosello<br>W John Edmunds |

The funders had no role in study design, data collection and interpretation, or the decision to submit the work for publication. For the purpose of Open Access, the authors have applied a CC BY public copyright license to any Author Accepted Manuscript version arising from this submission.

### Author contributions

James D Munday, Conceptualization, Data curation, Software, Formal analysis, Validation, Investigation, Visualization, Methodology, Writing – original draft, Project administration, Writing – review and editing, Conceived and designed the modelling framework and the evaluation of forecasts. Implemented the model and performed the formal forecast evaluations. Interpreted the results. Wrote and edited the manuscript; Alicia Rosello, Conceptualization, Data curation, Software, Formal analysis, Validation, Investigation, Visualization, Methodology, Writing – original draft, Project administration, Writing – review and editing, AR conceived and designed the interviews. Conducted the expert interviews and prepared the interview data for comparison. Interpreted the results. Wrote and edited the manuscript; W John Edmunds, Conceptualization, Data curation, Supervision, Funding acquisition, Validation, Methodology, Project administration, Writing – review and editing, Conceived and designed the interviews. Interpreted the results. Edited the manuscript; Sebastian Funk, Conceptualization, Data curation, Supervision, Funding acquisition, Validation, Methodology, Writing – original draft, Project administration, Writing – review and editing, Conceived and designed the modelling framework and the evaluation of forecasts. Interpreted the results. Edited the manuscript

### Author ORCIDs

James D Munday https://orcid.org/0000-0002-6206-7134
Alicia Rosello https://orcid.org/0000-0002-0737-5679
Sebastian Funk https://orcid.org/0000-0002-2842-3406

### Ethics

Human subjects: LSHTM ethics approval was obtained for this study (reference: 17633). Signed informed consent was taken from experts willing to participate and their verbal consent was requested again at the beginning of each elicitation.

Reviewer #1 (Public review): https://doi.org/10.7554/eLife.98005.3.sa1
Reviewer #2 (Public review): https://doi.org/10.7554/eLife.98005.3.sa2
Author response https://doi.org/10.7554/eLife.98005.3.sa3

# Additional files

## Supplementary files
MDAR checklist

## Data availability
All data and code used to process the expert interview responses can be found here: https://github.com/epiforecasts/Ebola-Expert-Interviews (copy archived at *Epiforecasts, 2025a*). The forecasts were performed using the EpiCastR package https://github.com/epiforecasts/EpiCastR (copy archived at *Epiforecasts, 2025b*). The code used for the analysis and scoring of the forecasts can be found here: https://github.com/epiforecasts/ebola-expert-elicitation (copy archived at *Epiforecasts, 2025c*).

The following previously published dataset was used:

| Author(s) | Year | Dataset title | Dataset URL | Database and Identifier |
|---|---|---|---|---|
| World Health Organization | 2019 | DRC Health Data Ebola 2018 | https://data.humdata.org/dataset/ebola-cases-and-deaths-drc-north-kivu | Humanitarian Data Exchange (HDX), ebola-cases-and-deaths-drc-north-kivu |

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
