## [Editor Report · eLife Assessment]

This article provides **valuable** evidence comparing the performance of mathematical models and opinions from experts engaged in outbreak response in forecasting the spatial spread of an Ebola epidemic. The evidence supporting the conclusions is **convincing**. It will be of interest to disease modellers, infectious disease epidemiologists, policymakers, and those who need to inform policymakers during an outbreak.

---

## [Referee Report · Reviewer #1 (Public review)]

Munday, Rosello, and colleagues compared predictions from a group of experts in epidemiology with predictions from two mathematical models on the question of how many Ebola cases would be reported in different geographical zones over the next month. Their study ran from November 2019 to March 2020 during the Ebola virus outbreak in Democratic Republic of the Congo. Their key result concerned predicted numbers of cases in a defined set of zones. They found that neither the ensemble of models nor the group of experts produced consistently better predictions. Similarly, neither model performed consistently better than the other, and no expert's predictions were consistently better than the others'. Experts were also able to specify other zones in which they expected to see cases in the next month. For this part of the analysis, experts consistently outperformed the models. In March, the final month of the analysis, the models' accuracy was lower than in other months, and consistently poorer than the experts' predictions.

A strength of the analysis is use of consistent methodology to elicit predictions from experts during an outbreak that can be compared to observations, and that are comparable to predictions from the models. Results were elicited for a specified group of zones, and experts were also able to suggest other zones that were expected to have diagnosed cases. This likely replicates the type of advice being sought by policymakers during an outbreak.

A potential weakness is that the authors included only two models in their ensemble. Ensembles of greater numbers of models might tend to produce better predictions. The authors do not address whether a greater number of models could outperform the experts.

The elicitation was performed in four months near the end of the outbreak. The authors address some of the implications of this. A potential challenge for the transferability of this result is that the experts' understanding of local idiosyncrasies in transmission may have improved over the course of the outbreak. The model did not have this improvement over time. The comparison of models to experts may therefore not be applicable to early stages of an outbreak when expert opinions may be less well-tuned.

This research has important implications for both researchers and policy-makers. Mathematical models produce clearly-described predictions that will later be compared to observed outcomes. When model predictions differ greatly from observations, this harms trust in the models, but alternative forms of prediction are seldom so clearly articulated or accurately assessed. If models are discredited without proper assessment of alternatives then we risk losing a valuable source of information that can help guide public health responses. From an academic perspective, this research can help to guide methods for combining expert opinion with model outputs, such as considering how experts can inform models' prior distributions and how model outputs can inform experts' opinions.

Comments on revisions:

I am grateful to the authors for their responses to my previous comments. I think their updates have made the paper much clearer. I do not think the updates change the opinions already given in the public review so I have not modified it.

---

## [Referee Report · Reviewer #2 (Public review)]

The manuscript by Munday et al. presents real-time predictions of geographic spread during an Ebola epidemic in north-eastern DRC. Predictions were elicited from individual experts engaged in outbreak response and from two mathematical models. The authors found comparable performance between experts and models overall, although the models outperformed experts in a few dimensions.

Both individual experts and mathematical models are commonly used to support outbreak response, but the relative strengths of each information source are rarely quantified. The manuscript presents an in-depth analysis of the accuracy and decision-relevance of the information provided by each source individually and in combination for a real-time outbreak response effort.

While this paper presents an important and unique comparison, forecast performance is known to be inconsistent and unpredictable across many dimensions such as pathogen, location, forecasting target, and phase of the outbreak. Thus, as the authors note, continuing to replicate such studies will be important for verifying the robustness of their conclusions in other contexts.

Comments on revisions:

I have no further comments. I commend the authors for an interesting and important contribution.

---

## [Author Response]

The following is the authors’ response to the original reviews.

**Reviewer #1 (Public review):**
Munday, Rosello, and colleagues compared predictions from a group of experts in epidemiology with predictions from two mathematical models on the question of how many Ebola cases would be reported in different geographical zones over the next month. Their study ran from November 2019 to March 2020 during the Ebola virus outbreak in the Democratic Republic of the Congo. Their key result concerned predicted numbers of cases in a defined set of zones. They found that neither the ensemble of models nor the group of experts produced consistently better predictions. Similarly, neither model performed consistently better than the other, and no expert's predictions were consistently better than the others. Experts were also able to specify other zones in which they expected to see cases in the next month. For this part of the analysis, experts consistently outperformed the models. In March, the final month of the analysis, the models' accuracy was lower than in other months and consistently poorer than the experts' predictions.A strength of the analysis is the use of consistent methodology to elicit predictions from experts during an outbreak that can be compared to observations, and that are comparable to predictions from the models. Results were elicited for a specified group of zones, and experts were also able to suggest other zones that were expected to have diagnosed cases. This likely replicates the type of advice being sought by policymakers during an outbreak.A potential weakness is that the authors included only two models in their ensemble. Ensembles of greater numbers of models might tend to produce better predictions. The authors do not address whether a greater number of models could outperform the experts.The elicitation was performed in four months near the end of the outbreak. The authors address some of the implications of this. A potential challenge to the transferability of this result is that the experts' understanding of local idiosyncrasies in transmission may have improved over the course of the outbreak. The model did not have this improvement over time. The comparison of models to experts may therefore not be applicable to the early stages of an outbreak when expert opinions may be less welltuned.This research has important implications for both researchers and policy-makers. Mathematical models produce clearly-described predictions that will later be compared to observed outcomes. When model predictions differ greatly from observations, this harms trust in the models, but alternative forms of prediction are seldom so clearly articulated or accurately assessed. If models are discredited without proper assessment of alternatives then we risk losing a valuable source of information that can help guide public health responses. From an academic perspective, this research can help to guide methods for combining expert opinion with model outputs, such as considering how experts can inform models' prior distributions and how model outputs can inform experts' opinions.
**Reviewer #2 (Public review):**
Summary:The manuscript by Munday et al. presents real-time predictions of geographic spread during an Ebola epidemic in north-eastern DRC. Predictions were elicited from individual experts engaged in outbreak response and from two mathematical models. The authors found comparable performance between experts and models overall, although the models outperformed experts in a few dimensions.Strengths:Both individual experts and mathematical models are commonly used to support outbreak response but rarely used together. The manuscript presents an in-depth analysis of the accuracy and decision-relevance of the information provided by each source individually and in combination.Weaknesses:A few minor methodological details are currently missing.

We thank the reviewers for taking the time to consider our paper and for their positive reflections and suggestions for our study. We recognise and endorse their characterisation of the study in the public reviews and are greatful for their interest and support for this work.

**Reviewer #1 (Recommendations For The Authors):**
I initially found Table 1 difficult to interpret. In the final two columns, the rows relate to each other but in the other columns, rows within months don't relate to each other. Could this be made clearer?

Thank you for your helpful suggestion. We agree that this is a little confusing and have now added vertical dividers to the table to indicate which parts of the table relate to each other.

In Figure 1A, the colours are the same as in the colour-bar for Figure 1B but don't have the same meaning. Could different colours be used or could Figure 1A have its own colour-bar to aid clarity?

Thank you for your query. The colours are not the same pallette, but we appreciate that they look very similar. To help the reader we have changed the colour palette of panel A and added a legend to the left.

In Figure 3, can labels for each expert be aligned horizontally, rather than moving above and below the timeline each month?

Thank you for your perspective on this. We made the concious dicision to desplay the experts in this way as it allows the timeline to be presented in a shorter horizontal space. We appreciate that others may prefer a different design, but we are happy with this one.

On lines 292 and 293, the authors state that experts were less confident that case numbers would cross higher thresholds. It seems that this would be inevitable given the number of cases is cumulative. Could this be clarified, please?

Thank you for raising this point. We agree that this wording is confusing. We have now reworked the entire section in response to another reviewer. The equivalent section now reads:

Experts correctly identified Mabalako as the highest-risk HZ in December. They attributed an average 82% probability of exceeding 2 cases; Mabalako reported 38 cases that month, exceeding all thresholds, although the probability assigned to exceeding the higher thresholds was similar to that of Beni (3 cases)

**Reviewer #2 (Recommendations For The Authors):**
(1) Some methodological details seem to be missing. Most importantly, the results present multiple ensembles (experts, models, and both), but I can't seem to find anywhere in the Methods that details how these ensembles are calculated. Also, I think it would be useful to define the variables in each equation. It would have been easier to connect the equations to the description if the variables were cited explicitly in the text.

Thank you for pointing out these omissions. We have included the following paragraph to detail how ensemble forecasts were calculated.

“Enslemble forecasts

Ensemble forecasts were calculated as an average of the probabilities attributed by the members of the ensemble. For the expert ensemble the arithmetic mean was calculated across all experts with equal weighting. Similarly the model ensemble used the unweighted mean of the model forecasts. For the mixed (model and expert) ensemble, the mean was weighted such that the combined weight of the experts forecasts and the combined weight of the models forecasts were equal.”

(2) Overall, I think the results provide a strong analysis of model vs. expert performance. However, some sections were highly detailed (e.g., the text usually discusses results for every month and all health zones), which clouded my ability to see the salient points. For example, I found it difficult to follow all the details about expert/model predictions vs. observations in the "Expert panel and health zones..." subsection; instead, the graphical illustration of predictions vs. observations in Figure 4 was much easier to interpret. Perhaps some of these details could be trimmed or moved to the supplementary material.

Thank you for your honest feedback on this point. We have shortened this section to highlight the key points that we feel are the most important. We have also simplified the text where we discuss the health zones nominated by experts.

(3) Figure 5C is a nice visualization of the fallibility of relying on a single individual expert (or model). I wonder if it would be useful to summarize these results into the probability that a randomly selected expert outperforms a single model. Is it the case that a single expert is more unreliable than a single model? The discussion emphasizes the importance of ensembles and compares a single model to an ensemble of experts, but eliciting predictions from multiple experts may not always be possible.

Thank you for raising this. We agree that this is an important point that eliciting expert opinions is not a trivial task and should not be taken for granted. We agree with the principle of your suggestion that it would be useful to understand how the models compare to indevidual experts. We don’t however believe that an additional analysis would add sufficiently more information than already shown in Figure 5, which already displays the full distribution of indevidual experts for each month and threshold. If you would like to try this analysis yourself, the relevant data (the indevidual score for each combination of expert, threshold, heal zone and month) is included in the github repo (https://github.com/epiforecasts/Ebola-Expert-Elicitation/blob/main/outputs/indevidual_results_with_scores.csv).

Minor comments:(1) Figure 2: the color scales in each panel are meant to represent different places, correct? The figure might be easier to interpret if the colors used were different.

Thank you for bringing this to our attention. We have now changed the palette of panel A to differ from panel B.

(2) Equation 7: is o(c>c_thresh) meant to be the indicator function (i.e. 1 if c>c_thresh) and 0 otherwise?

Thanks for raising this. The function o is the same as in the previous equation – an observation count function. We appreciate that this is not immediately clear so have added a sentence to explain the notation after the equation.

(3) Table 1: a brief description of the column headers would be useful.

Thank you for the suggestion. We have now extended the table caption to include more description of the columns.

“Table 1: Experts and health zones included in each round of the survey. The left part of the table details the experts interviewed (highlighted in green) the health zones included in the main survey in each month. In addition, the right part of the table details the health zones nominated by experts and the number of experts that nominated each one.”